# Promising Diagnostic and Therapeutic Approaches Based on VHHs for Cancer Management

**DOI:** 10.3390/cancers16020371

**Published:** 2024-01-15

**Authors:** Ying Cong, Nick Devoogdt, Philippe Lambin, Ludwig J. Dubois, Ala Yaromina

**Affiliations:** 1The M-Lab, Department of Precision Medicine, GROW—School for Oncology and Reproduction, Maastricht University, 6211 LK Maastricht, The Netherlands; y.cong@maastrichtuniversity.nl (Y.C.); philippe.lambin@maastrichtuniversity.nl (P.L.); 2Molecular Imaging and Therapy Research Group (MITH), Vrije Universiteit Brussel, 1090 Brussels, Belgium; ndevoogd@vub.be; 3Department of Radiology and Nuclear Medicine, Maastricht University Medical Centre+, 6229 HX Maastricht, The Netherlands

**Keywords:** VHH, nanobody, single domain antibody, cancer, therapy, diagnosis

## Abstract

**Simple Summary:**

With around 15 kD molecular weight, VHHs are considered the smallest antigen-binding fragments. Based on their unique properties, VHHs have broad application prospects such as bio-sensing, molecular imaging, drug delivery, disease treatments, and diagnosis. In this review, we discuss VHH applications for cancer treatments and diagnosis. Meanwhile, an overview of VHH-based agents in clinical trials is provided.

**Abstract:**

The discovery of the distinctive structure of heavy chain-only antibodies in species belonging to the Camelidae family has elicited significant interest in their variable antigen binding domain (VHH) and gained attention for various applications, such as cancer diagnosis and treatment. This article presents an overview of the characteristics, advantages, and disadvantages of VHHs as compared to conventional antibodies, and their usage in diverse applications. The singular properties of VHHs are explained, and several strategies that can augment their utility are outlined. The preclinical studies illustrating the diagnostic and therapeutic efficacy of distinct VHHs in diverse formats against solid cancers are summarized, and an overview of the clinical trials assessing VHH-based agents in oncology is provided. These investigations demonstrate the enormous potential of VHHs for medical research and healthcare.

## 1. Introduction

In 1993, a different type of antibody called heavy chain only antibody (HCAb) was discovered in camelids, which had another structure compared with conventional antibodies (Figure 1) [1]. Another type of antibody lacking light chains is also found in cartilaginous fish [2,3]. HCAbs only contain two heavy chains, each one bearing beside two constant regions CH2 and CH3 also a variable region. This variable region is also called the variable domain of heavy chain of heavy-chain-only antibody (VHH), can be recombinantly produced and has a molecular weight of only about 15 kD, which is 10% of a conventional antibody and half of a single-chain variable fragment (scFv). Therefore, VHHs are also called single domain antibodies (sdAb) or nanobodies.

While conventional antibodies have been used in anti-cancer therapies for many years, VHH-based therapies have more recently started to be explored. The unique features of VHH, such as high affinity and lower immunogenicity, may provide important differences in anti-cancer therapies. To date, preclinical studies and early-stage clinical trials have demonstrated the promising potential of VHH for cancer management. In this review, we provide an overview of VHH characteristics and the general route of VHH development. Furthermore, we summarize the applications of VHH in anti-cancer therapies and provide an overview of registered clinical trials testing VHH-based therapeutic and diagnostic agents for oncological diseases.

## 2. General Characteristics of VHH

In VHH, there are only three complementarity-determining regions (CDRs) that make contact with the antigen, as compared to six in conventional antibodies (3 in VH and 3 in VL). To compensate for this and to ensure sufficient contact interface, the amino chain in CDR3 (and to a lesser extent CDR1) is longer in VHH than its counterpart in conventional antibodies [4,5]. As a result, the diversity and specificity of CDR3 is improved. Additionally, CDR3 in VHH can make up an exposed ring structure, which can behave like a ‘finger’ to stick into the ‘pocket’ of antigen, where a conventional antibody rather interacts with flat surfaces [6,7,8,9]. In addition, an extra cysteine residue in CDR3 can form a disulfide bond with an extra cysteine residue in either CDR1 or in the framework region 2 (FR2), which increases the stability of the VHH structure, while decreasing the energy requested for binding with antigens [4,5].

### 2.1. Biochemical and Biophysical Characteristics

The unique biochemical and biophysical characteristics of VHH are their stability, in addition to high solubility, thermo- and proteolytic-resistance [10]. FR2 contains many hydrophilic amino acid residues, which determines the high solubility and dispersibility of VHH in water. VHH has also good conformational and thermal stability, which allows VHH to be stable at body temperature during a week while still having sufficient binding ability [11]. Additionally, VHH is also stable at extreme pH conditions, ensuring that VHH maintains its bioactivity in the stomach or the intestine [12,13,14,15]. The stability of VHH makes it possible to design VHH treatments following various administration methods, including intravenous injection, inhalation, and oral and intranasal administration. Altogether, the stable biochemical and biophysical properties nurture the increasing applications of VHH.

### 2.2. Low Immunogenicity

Relatively little scientific evidence is publicly available to make strong claims about the immunogenicity of VHHs. Nevertheless, the structure of VHH has a high similarity with the human VH structure, in particular, it has a high homology (86–94%) with the human VH3 family [16,17]. The relatively small size of VHH (125 amino acids) implies a low number of potentially immunogenic epitopes. The high solubility of VHHs reduces the likeliness to form highly immunogenic aggregates. Moreover, unmodified VHHs show fast blood clearance, which minimizes capture by antigen-presenting cells. All these features favor low immunogenicity. In a phase I clinical study of ^68^Ga-HER2-VHH, 20 patients were free of anti-drug antibodies (ADA) after one injection [18]. A phase I study of ALX-0141, a trimeric format consisting of two identical anti-RANKL humanized VHHs and one anti-albumin humanized VHH, demonstrated that after a single subcutaneous injection in 42 healthy volunteers, no ADAs were found [19]. In summary, the immunogenicity of VHH in humans seems to be relatively low, which prevents one of the main causes of adverse effects, i.e., allergy. Humanization of VHH is possible and can further reduce its immunogenicity [20,21,22].

### 2.3. High Tissue Penetration and Fast Blood Clearance

Theoretically, the passive diffusion rate of a molecule in tissue is inversely proportional to its molecular size. Therefore, compared with conventional antibody (150 kD), the monovalent VHH (15 kD) has a fast blood extravasation and a better tissue penetration to reach its target. This results in a more homogeneous distribution, for example in solid tumors, compared to conventional antibodies [23,24,25,26]. It has been shown that VHHs can cross the blood-brain barrier (BBB) of healthy rats [27], providing improved ability for the diagnosis and treatment of brain cancer, where in addition the BBB is likely disrupted [28,29,30,31,32]. Due to the small molecular weight, VHH is usually cleared quickly through the kidneys [33,34,35,36]. As a result, the half-life time of VHH in the human body is mostly less than an hour [37,38], a feature helpful to prevent toxicity. Although the short half-life of VHH is advantageous for diagnostic applications, it may limit its therapeutic efficacy [39,40,41]. Methods such as PEGylation, fusion or binding with albumin, multimerization, or fusion with antibody Fc can help to extend the half-life [42,43,44,45,46,47]. For example, a bivalent anti-VEGF VHH showed a 1.8-fold longer half-life in comparison with the monovalent VHH [42]. PEGylation resulted in a 12-fold longer half-life of an anti-CEA/CD3 bispecific VHH [45].

## 3. The Generation of VHH

There are several steps in the identification of a functional VHH.

The construction of a VHH gene library

The first step in the generation of antigen-specific VHHs is to construct a VHH-encoding cDNA library. Antigen-specific VHHs can be retrieved either from an immune VHH library, a naive VHH library, or a synthetic VHH library. The immune library is the most frequently used method, exploiting the in vivo affinity maturation of the VHHs. To construct an immune library, camelid animals are immunized with a targeted antigen. The targeted antigen is injected into camelids multiple times, e.g., 7 times every two weeks [48]. 5~7 days after the immunization lymphocytes are then extracted from the peripheral blood, which contains B cells producing the matured HCAbs. Lymphocyte RNA is extracted and reverse transcribed into cDNA [49]. The bulk of cDNA is used as a template for PCR with VHH-specific primers [50]. Subsequently, the amplified VHH DNA fragments are cloned into bacterial vectors to construct the library [51].

2.The selection of specific VHH

Phage display is the dominant technique to screen libraries for functional VHHs. Here, the VHH immune libraries are cloned in phagemids in fusion with bacteriophage gene III, and then VHHs are displayed on the tip of the M13 bacteriophage by infection with helper phages providing phage structural components in trans. The target antigen is fixed directly or indirectly on a solid carrier, to allow the phage library to interact with the antigen. While the phages of interest bind with the fixed antigen, unbounded or unspecific phages are removed by washing. Subsequently, a specific acid (e.g., 0.1 M pH 2.2 HCl-Glycine) or base (e.g., 0.1 M triethylamine) is used to release the phage from its antigen. Generally, after 2–5 selection rounds of panning, the VHHs with good specificity and affinity can be selected [52].

3.The production of VHH

Due to its simple structure and good water solubility, VHH can be mass-produced by using relatively low-cost expression systems such as bacterial or yeast expression systems [51,53,54,55,56,57]. After verifying the nucleic acid sequence of the selected VHH, an expression vector is constructed and transferred into bacteria (e.g., *E. coli* BL21) or yeast (e.g., Pichia pastoris GS115 or X33) for the production of VHH. In the bacteria expression process, successfully transformed colonies are picked for culture expansion and induction of VHH production. For example, if a Lac-operon is present in the expression vector, IPTG (Isopropyl-β-d-thiogalactoside) is used to induce gene expression when the bacteria reach the exponential growth phase. Most often VHHs are transported to the periplasm and extracted from there, although also cytoplasmic expression is being pursued [58]. In the yeast expression process, successfully transformed yeast colonies are picked for culture expansion following induction, with VHHs most often ending up in the conditioned medium due to secretion. The VHH protein in the supernatant, cell lysates, or periplasmic extracts can be purified by methods such as affinity chromatography, ion exchange chromatography, or molecular sieve chromatography. The production of VHHs in mammalian cell lines is rarely reported because of the high cost of this expression system.

## 4. Applications of VHH

### 4.1. Molecular Imaging

Molecular imaging is one of the important methods in cancer research, which allows the noninvasive study of tumors and their microenvironment, as well as tracing or monitoring cancer progression and therapy effectiveness. Generally, a molecular targeted imaging agent contains a targeting moiety and a radioisotope for nuclear imaging or a fluorescent moiety for optical imaging [59]. To target cancer cells more precisely, the imaging agent is required to have higher specificity and sensitivity, with lower toxicity and fewer adverse effects. 

Conventional antibodies have been frequently tested in cancer molecular imaging studies [60]. However, the application of conventional antibody-based molecular imaging is rather limited by its large size and relatively long plasma half-life. This requires the use of radionuclides with a long physical half-life, resulting in more radiation exposure and additional measures for radiation protection [61]. With their smaller molecular weight, VHHs have a deeper penetration into the central part of solid tumors to detect target expression [62,63]. In addition, the short half-life and fast blood clearance limit non-specific VHH presence resulting in an improved signal-to-noise ratio and fewer adverse effects. Therefore, the VHH-based imaging technique is currently heavily investigated for cancer diagnosis, to target proteins or receptors which are overexpressed in tumor cells or the tumor microenvironment during cancer progression.

Positron emission tomography (PET) and single photon emission computed tomography (SPECT) are the two main nuclear imaging techniques for cancer diagnosis. To improve the accuracy and specificity and to reduce potential side effects of radiation exposure in PET/SPECT imaging, the VHH has been used as an ‘immune carrier’ in the construction of radioactive probes. In the very first study, Gainkam et al. investigated the uptake of the ^99m^Tc labeled anti-EGFR VHH and showed that tumor uptake was correlated with tumor burden, concluding that the VHH can be used to track the tumor response to therapy [34]. Table 1 identifies several preclinical studies on VHH-radionuclide conjugates for imaging. Clinically, in a phase II study, ^68^Ga labeled anti-HER2 VHH was used to assess HER2 expression in primary breast cancer and metastatic lesions [64]. This clinical study showed that the ^68^Ga-HER2-nanobody could be safely administered, with favorable biodistribution and high accumulation in HER2-positive primary breast cancer and its metastases. In another recent phase I study, a ^68^Ga-labeled VHH against the macrophage mannose receptor CD206 was tested in a phase I PET safety study to track tumor-associated macrophages in the tumor stroma [65]. Other clinical studies are summarized in Table 7.

Besides being conjugated with a radionuclide, VHH can also be conjugated with a fluorescent moiety for molecular imaging [85]. For example, Debie et al. extensively evaluated fluorescently labeled anti-HER2 VHH for image-guided surgery and demonstrated a significant reduction of residual tumor lesions as compared to conventional surgery [86,87]. Van Brussel et al. developed a CAIX-specific VHH conjugated to IRDye800CW, that could visualize tumors compared to the background 2–3 h after the anti-CAIX VHH injection [88]. Furthermore, the VHH can also be used in magnetic resonance imaging (MRI) by conjugation with MRI contrast agents, such as gadolinium or paramagnetic nanoparticles [89,90].

### 4.2. VHHs in Anti-Cancer Therapies

The unique features that have been discussed above make VHHs promising replacements for monoclonal antibodies or other types of antibody fragments. Figure 2 generally describes different applications of VHHs in anti-cancer therapies. 

#### 4.2.1. Radioimmunotherapy

Radioimmunotherapy (RIT) is a “targeted” radionuclide therapy coupling radioisotopes with therapeutic properties to antibodies. These antibodies bind to specific targets in the tumors and deliver their cargo, i.e., the radionuclide, at that site, resulting in minimal toxicity to normal tissue [91]. ^90^Y-ibritumomab tiuxetan (Zevalin^®^, Bayer) and ^131^I-tositumomab (Bexxar^®^, GSK) are so far the only two RITs approved for clinical use to treat hematological malignancies [92,93]. Both are designed based on anti-CD20 monoclonal antibodies to treat CD20-positive non-Hodgkin’s lymphoma (NHL). The application of RIT for solid tumor treatment is, however, still a challenge. In addition to the large size of conventional antibodies, the leakage of tumor blood vessels and the complexity of the matrix in solid tumors make it difficult for RIT to reach all tumor cells. Therefore, alternative carriers of therapeutic radioisotopes, such as VHHs and peptides have been investigated for cancer RIT, especially for the treatment of solid tumors.

Table 2 summarizes the preclinical studies on the application of VHH in RIT of solid tumors. For example, D’Huyvetter et al. treated mice bearing HER2-positive SKOV3 ovarian adenocarcinoma xenografts with the ^177^Lu-labeled anti-HER2 VHH and demonstrated that the tumor growth was almost completely inhibited. Moreover, the event-free survival was significantly longer for the treated group compared to the control group that received vehicle treatment, while no evidence of renal inflammation or necrosis was observed [94]. Similarly, anti-FAP VHHs that are conjugated to ^131^I or ^225^Ac were able to limit tumor growth and enhance the survival of mice bearing FAP-expressing tumors [95]. To target the TME, Xu et al. generated the anti-FAPα (fibroblast activation protein-α) VHH-Fc fusion labeled with ^177^Lu and showed therapeutic efficacy in HT1080 fibrosarcoma xenografts in mice [96]. Anti-MMR VHH labeled with ^177^Lu was designed to target the stroma of solid tumors. The ^177^Lu-labeled anti-MMR VHH significantly delayed tumor growth, which outcompeted the effects of currently used therapies, such as immune checkpoint inhibitors, anti-angiogenic therapy (anti-VEGFR2 mAbs), and chemotherapies (doxorubicin, paclitaxel) [97].

#### 4.2.2. Photodynamic Therapy

VHH can be conjugated to a photosensitizer (PS), which is a light-activatable compound that generates cytotoxic reactive oxygen species upon activation with a certain light wavelength [104]. These VHH-PS conjugates can also be used as tracers or probes to monitor tumor progression using optical imaging after photodynamic therapy (PDT). IRDye700DX, a phthalocyanine dye, is one of the most promising photosensitizers, whose chemical structure is similar to hematoporphyrin. Its maximum absorption wavelength is in the infrared wavelength region and therefore easier to be transmitted through human tissue. Additionally, IRDye has low absorption of light with a wavelength of 400–600 nm, which reduces the skin photosensitivity reaction. Therefore, many studies have used IRDye700DX as PS to couple with VHH for PDT. For example, Renard et al. generated anti-EGFR VHH conjugated with ^111^In-DPTA (diethylenetriaminepentaacetic acid) and IRDye700DX PS in a site-specific way, which homed to A431 xenografts in vivo and resulted in light-induced toxicity via cellular internalization [105]. Deken et al. investigated anti-HER2 VHH conjugated with IRDye700DX PS that induced significant regression of trastuzumab-resistant HER2-expressing breast tumors after a single treatment session with selective cytotoxicity [106]. Table 3 summarizes cancer targets investigated using VHH-PS conjugates in preclinical studies.

#### 4.2.3. VHH as Immune Checkpoint Inhibitor

Several conventional antibodies have been used as immune checkpoint inhibitors (ICIs) for anti-cancer treatments in the past years. However, due to the complex structure of these antibodies and the tumor heterogeneity, antibody-based ICIs face many challenges such as insufficient tumor penetration and immune-related adverse effects [112], highlighting the need for optimization of antibody-based ICIs.

Due to the smaller molecular weight, VHHs have a higher penetration potential in solid tumors. Petit et al. showed that T cell-mediated targeted delivery of an anti-PD-L1 VHH outperformed an anti-PD-L1 conventional antibody in inhibiting tumor growth, related to its higher tumor penetration in MC38 tumor-bearing mice [113]. Table 4 summarizes the studies evaluating VHH-based ICIs. Although the VHHs lack ADCC and CDC due to the absence of Fc fragment, the VHH still has anti-cancer efficacy by inhibiting the bioactivity of immune checkpoints to attenuate immune suppression in tumors [114]. Therefore, fusing VHH with an Fc fragment can also compensate the ADCC and CDC in anti-cancer therapies [115]. Ma et al. demonstrated that a tetravalent anti-PD-L1 VHH-Fc fusion had a higher inhibitory effect on tumor growth compared to an equimolar dose of an anti-PD-L1 monoclonal antibody (Tecentriq analog) in MC38 xenograft-bearing mice [116]. Clinically, envafolimab (KN035), anti-PD-L1 VHH-Fc fusion formulated for subcutaneous injection, has demonstrated a favorable safety and pharmacokinetic profile, with promising antitumor activity in patients with advanced solid tumors in Phase I–II trials [117,118,119]. 

#### 4.2.4. Targeting Tumor-Specific Antigens

Tumor-specific antigens (TSAs) are important targets for cancer treatments and diagnosis. However, TSAs are not exclusively expressed in cancer cells. Normal cells may express TSAs at low levels, while cancer cells have high expression of TSAs during tumor proliferation. As a consequence, molecules targeting TSAs should have high specificity and affinity, which can be achieved with VHH.

VHHs have been designed to bind tumor-specific targets TSAs such as HER2, EGFR, and VEGFR, over the past years and they are reported to have anti-tumor efficacy. Table 5 summarizes the preclinical studies evaluating the therapeutic efficacy of various anti-TSA VHHs. As the small molecular weight of VHH allows better intratumor penetration, it leads to high VHH clearance resulting in a shorter plasma half-life during the treatment. In vivo studies showed the half-life of VHH binding with its target within 1–2 h [128,129]. In addition, VHHs are not capable of promoting ADCC or CDC effects during the treatment, which should also be considered. These two limitations can be overcome by modification of the VHHs. For example, Sadeghi et al. generated an anti-VEGF bivalent VHH that showed a 1.8-fold longer half-life compared to its monovalent form [42]. An et al. coupled ABD035 to anti-GPC3 VHH labeled with ^68^Ga, which reached a peak tumor uptake at 6 h post-injection combined with a reduced kidney accumulation at 1 h post-injection [78]. Another method is coupling VHHs with an Fc fragment to obtain the VHH-conjugates with the ability to promote ADCC or CDC [130]. 

#### 4.2.5. VHH-Drug Conjugates (VHH-DC)

Antibody-drug conjugates (ADCs) are immune-conjugates formed with a monoclonal antibody as a ‘delivery carrier’ and a cytotoxic/immune-promoting drug as ‘cargo’, which are conjugated via a chemical linker [144]. Highly specific targeting antibodies are used to overcome the toxicity and adverse effects of traditional chemotherapeutic drugs. In addition, the antibody may also have additive or synergistic anti-cancer effects with the drug, when both components of ADC exhibit anti-tumor effects [145]. Nowadays, third-generation ADCs have been developed and investigated. With novel techniques for ADC generation, ADCs with high targeting and therapeutic efficacy, relatively low immunogenicity, consistent drug-to-antibody ratio (DAR) and low toxicity are becoming available [146]. As more ADCs have been entering clinical trials, more challenges for ADC development have been raised. During the synthesis of ADC, the chemical modification leads to the instability of monoclonal antibodies. In addition, the high molecular weight makes it difficult for ADC to enter the inner hypoxic part of solid tumors, which limits the application of ADCs in cancer treatments. Using VHHs as a ‘carrier’ of cytotoxic drugs is a novel alternative to overcome the disadvantages of conventional monoclonal antibodies. Table 6 summarizes recent VHH-drug conjugates studies. Indeed, Wu et al. demonstrated that the anti-5T4 VHH-SN38 (irinotecan analog) conjugate was detected at 85 μm from the periphery of the patient-derived organoids (PDOs), while the anti-5T4 conventional antibody-SN38 conjugate mainly located around the periphery of PDOs [147]. Drug resistance and downregulation of tumor antigens are challenging for ADCs as well [148]. Espelin et al. used anti-HER2 VHH conjugated with doxorubicin (MM-302) in combination with trastuzumab to demonstrate synergistic anti-tumor activity both in vitro and in vivo, supporting its translation into clinical trials [149]. Therefore, other strategies are being investigated including utilization of novel tumor antigens, antibody formats, linkers, payloads, etc.

#### 4.2.6. VHH-Based CAR-T

The chimeric antigen receptor (CAR) is an engineered synthetic receptor that can redirect lymphocytes to recognize and eliminate specific cells, which express the target antigens [153]. T cells are most commonly used to carry CARs, but CARs can also be applied to other types of immune cells, such as NK cells, dendritic cells, macrophages, etc. [154,155]. Hitherto, six CAR-T therapies have been approved by the FDA to treat relapsed/refractory multiple myeloma (RRMM), diffuse large B-cell lymphoma (DLBCL), mantle cell lymphoma (MCL), follicular lymphoma (FL) and Precursor B-cell lymphoblastic leukemia, of which anti-CD19 CAR-T products’ antigen recognition domain is based on the same scFv (Kymriah^®^, Yescarta^®^, Tecartus^®^, and Breyanzi^®^), whereas the antigen recognition domain of B-cell maturation antigen (BCMA, Carvykti^®^, Legend Biotech and Janssen Biotech) is based on VHH [156]. The peptide linker of scFv-based CAR-T, however, showed immunogenicity as neutralizing antibodies were generated against the linker [157]. Additionally, T cell exhaustion is another factor that limits the application of CAR-T, which occurs independently of the binding with the target antigen but is caused by the unstable structure of scFv. The exposed hydrophobic residues in the scFv variable domain and poor VH or VL folding stability can result in CAR aggregation. The CAR aggregation on the surface of CAR-T can lead in turn to the activation of effector cells and cytotoxic signaling cascade, which contribute to T cell exhaustion [158,159]. Xie et al. generated VHH-based CAR-T cells, which were designed to target PD-L1, CD47, or EIIIB+ fibronectin splice variant to target the TME, and each VHH-based CAR-T could reduce solid tumor growth and improve survival in immunocompetent tumor-bearing mice [160,161]. Rajabzadeh et al. designed an anti-MUC1 CAR-T which could increase secretion of Th1 cytokines and cytotoxic activity that could inhibit cancer cell viability [162]. In summary, VHH-based CAR-T is promising as an approach to treating solid tumors [163].

### 4.3. Other Applications of VHHs

VHHs can also be applied in other novel anti-cancer treatments. VHH, which binds and neutralizes cytokines or growth factors such as TNFα or VEGF, respectively, can attenuate tumor growth by inhibiting metastasis or angiogenesis [13,164,165,166]. Alternatively, VHHs conjugated with proteins (e.g., cucurmosin, pseudomonas exotoxin), cytokines (e.g., IFNγ, IL-2, TNFα) or peptides show promising potential of inhibiting tumor growth [167,168,169,170,171]. Yin et al. investigated an anti-PD-L1 VHH and used it for modified liposomes co-delivery of simevastatin/gefitinib that could remodel the TME of EGFR-T790M-mutated NSCLC and overcome the drug resistance [172]. Furthermore, VHHs can be used to construct cancer vaccinations. For example, conjugating anti-CD11b VHH with H-2Db-restricted immunodominant E7 epitope (E749-57), which exists in high-risk human papillomavirus-associated cancers, induced stronger CD8^+^ T-cell responses against HPV positive tumors than E749-57 alone [173]. The tumor vaccine that contains anti-CD47 VHH could induce suppression of tumor progression and improve the long-term survival of tumor-bearing mice by remodeling the TME [174]. Altogether, VHHs have great potential to develop novel anti-cancer therapies with higher efficacy and lower toxicity, which warrants further investigations.

## 5. An Overview of Ongoing Clinical Trials of VHHs in Cancer Treatments

With the failure of many promising anti-cancer therapies in clinical trials and understanding the progression of cancer, more strict safety and efficacy requirements have been raised for the investigation of cancer treatments. Since the discovery of VHHs in 1993, VHH have become a promising agent for the diagnosis and treatment of cancer, viral infection [41,175,176], and other diseases due to its unique structure and physicochemical properties. The CAR-T, in which anti-BCMA VHHs serve as antigen receptors on T cells (Carvykti^®^, Legend Biotech, and Janssen Biotech), has been approved by the FDA for the treatment of advanced multiple myeloma in 2022. Meanwhile, many clinical trials evaluating the diagnostic and therapeutic potential of VHH-based agents in solid and hematological malignancies are ongoing. Table 7 summarizes some of these clinical trials. Data was collected on ClinicalTrial.gov, on 24 August 2023 (https://www.clinicaltrials.gov/ct2/home, accessed on 24 August 2023, Keywords: VHH, nanobody, sdAb, cancer, tumor, malignancy).

## 6. Discussion and Future Directions

Since its discovery in 1993, VHHs have become one of the most promising approaches for the treatment and diagnosis of various diseases, including cancer, autoimmune diseases, respiratory diseases, and blood systemic diseases. The first VHH-based drug Caplacizumab was approved in 2018 for the treatment of thrombotic thrombocytopenic purpura and thrombosis. In 2022, the first VHH-based CAR-T treatment Carvykti has been approved by the FDA to treat multiple myeloma. Up until now, VHH alone as a therapeutical agent has not been approved for anti-cancer treatments.

As the smallest known antibody fragment that can recognize antigen, the VHHs retain high affinity and specificity, whilst having unique chemical and physical features compared to conventional antibodies. The unique molecular structure of VHHs makes it possible to bind some epitopes that conventional antibodies are not able to access, such as the active structure in the middle of the protein cleft. These unique features may behave like a ‘double-edged sword’ in the therapeutic application of VHHs. For example, the single domain character of VHH makes almost every amino acid residue crucial for antigen-antibody interaction increasing the difficulty of VHH humanization. Additionally, fast blood clearance can reduce the toxicity of the treatment, but it may compromise treatment efficacy as well. In addition, although the lack of Fc fragments minimize immunogenicity, the absence of ADCC or CDC limits the application of VHHs in anti-cancer treatments, which are mainly promoted by Fc fragments. Therefore, related technologies should be applied in the VHHs development process to overcome these drawbacks of VHHs. In addition, in preclinical studies evaluating the efficacy of VHHs, the cancer models should be chosen carefully, and the data should be interpreted with caution, since in some studies VHH imaging agents recognizing human antigen were tested in human tumor xenografts in mice, which cannot provide information on the binding of VHH in normal tissues of human origin. Similarly, VHH with immune-modulating properties targeting human antigens should be investigated in humanized mouse models.

## 7. Conclusions

According to the database of clinical trials investigating VHH-based anti-cancer agents, most VHHs in ongoing clinical trials are still in their early stages. Most of these anti-cancer targets cover PD-1, PD-L1, CTLA4, HER2, EGFR, and BCMA. The VHH studies gathered in this review indicate that many promising novel VHH-based anti-cancer treatments and diagnostic tools have been investigated. With the progressive understanding of cancer biology, more potential anti-cancer targets have been identified. Together with straightforward selection methods, efficient and cost-effective expression, and purification technological processes, VHHs represent a promising opportunity in anti-cancer treatment and cancer diagnosis. 

## Figures and Tables

**Figure 1 cancers-16-00371-f001:**
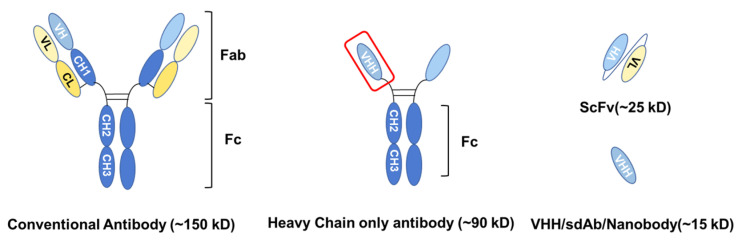
General structural comparison between conventional antibodies, heavy chain-only antibodies, single-chain Fv, and VHH. Abbreviations: Fab—Fragment of Antigen Binding; Fc—Fragment Crystallizable; CH—Constant domain of the Heavy chain; single-chain Fv; VH—Variable domain of the Heavy chain; CL—Constant domain of the Light chain; VL—Variable domain of the Light chain; VHH—Variable domain of the Heavy chain of Heavy-chain antibody.

**Figure 2 cancers-16-00371-f002:**
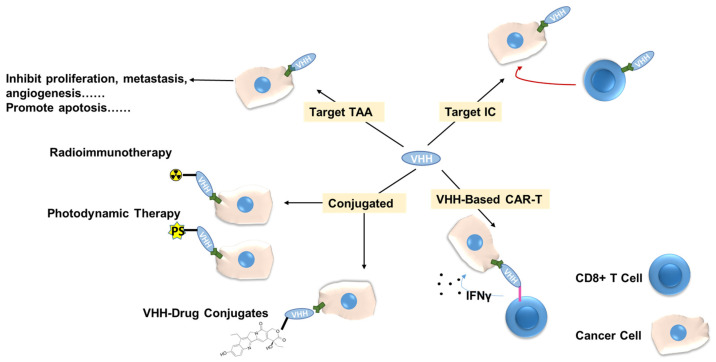
A brief description of VHHs in anti-cancer therapies. TAA—Tumor Associated Antigen, PS—photosensitizer, IC—Immune Checkpoint.

**Table 1 cancers-16-00371-t001:** Preclinical studies using VHH-radionuclide conjugates for imaging.

Target	Conjugation	Cancer Models	Main Findings	Reference
CD8	^89^Zr	BrCa	The CD8^+^ T cells in solid tumors were monitored by ^89^Zr-labeled anti-CD8-VHH, which signal positively corresponded with ICI treatment response.	[66]
CD8	^18^F	ALL	Imaging with the ^18^F-VHH enabled rapid visualization of CD8^+^ T cells within 1 h, while no visible tumor uptake was observed with the control VHH.	[67]
PSMA	^111^In	CRPC	Renal uptake was efficiently reduced by co-injection of gelofusine and lysine. Replacing the c-myc-his tag with the cysteine reduced renal uptake without loss of targeting.	[68]
PD-L1	^99m^Tc	TC-1 (immortalized murine lung epithelial cell)	VHH accumulation correlated with the levels of PD-L1 in tumors, even if PD-L1 expression was low.	[26]
PD-L1	^99m^Tc	NSCLC	[^99m^Tc]Tc-HYNIC-KN035 displayed a high PD-L1 specificity both in vitro and in vivo, that was positively correlated with the expression of PD-L1.	[69]
PD-L1	^68^Ga	SKCM, BrCa	^68^Ga-NOTA-Nb109 specifically accumulated in tumors with a maximum uptake of 5 ± 0.35% injected dose/g at 1 h.	[70]
PD-L1	^68^Ga	GBM, CRC, NSCLC	Tumor-to-muscle ratio (TMR) reached its peak at 40 min post-injection. The heart uptake was almost fully cleared at 35 min post-injection.	[71]
MMR	^99m^Tc	TS/A (murine mammary adenocarcinoma), 3LL-R (Lewis Lung carcinoma)	Anti-MMR VHH targeted pro-angiogenic MMR-expressing TAMs with tumor uptake correlating with the amount of TAMs in the tumor.	[72]
MMR	^68^Ga	3LL-R	TMR was determined while no treatment-related toxicologically relevant changes or acute immunological reactions were observed. The tolerated dose was established to be >1.68 mg/kg body weight. The dosimetry levels for humans were calculated by using the data in mice.	[73]
LAG-3	^99m^Tc	TC-1 (immortalized murine lung epithelial cell)	The tumor uptake of VHHs 3132 and 3206 targeting LAG-3 was comparable with high contrast at 1 h post-injection.	[74]
LAG-3	^99m^Tc	MC38 (murine CRC), MO4 (murine melanoma), and TC-1	The radiolabeled anti-LAG-3 VHH detected LAG-3 expressing TILs 1 h post tracer injection.	[75]
HER2	^68^Ga	HER2^+^ cancer	A high tumor-to-organ ratio was measured at 1 h post-injection with increased uptake upon increasing the injected dose.	[76]
HER2	^18^F	OvCa	The tumor-to-organ ratio at 1 h post-injection showed excellent specificity.	[25]
HER2	^99m^Tc	BrCa	The tumor had significant radiotracer uptake at 0.5 h after injection.	[77]
Glypican-3	^68^Ga, ^18^F	HCC	The fusion of VHH to an albumin-binding domain increased the tumor uptake and decreased kidney accumulation of the radiotracer (1 h to 6 h).	[78]
EpCAM	^99m^Tc	EpCAM driven cancer	The uptake value in tumors was increased about two times from 0.5 h till 12 h after injection, while it could clearly image tumor-draining lymph nodes.	[79]
EGFR	^99m^Tc	EGFR^+^ cancer, A431 (epidermoid carcinoma)	VHH uptake correlated with tumor burden and tumor response to EGFR inhibitor (erlotinib).	[34]
EGFR	^99m^Tc	A431	In vivo, the study demonstrated that OA-cb6 labeled with ^99m^Tc showed an approximately 2.7-fold tumor-muscle ratio at 4 h post-injection.	[80]
CLDN18.2	^89^Zr	STAD	The VHH had good tumor uptake to evaluate the expression of CLDN18.2 in gastric cancer for patient selection.	[81]
CEACAM5	^99m^Tc	NSCLC	The high ratio of the signal in the tumor compared with the background confirmed that the VHH can be used as a molecular probe for imaging CEACAM5-expressing tumors.	[82]
CAIX	^111^In	HNSCC	The anti-CAIX VHH targeted hypoxia regions in solid tumors.	[83]
* EDB of FN	^64^Cu	pan-cancer	Targeted the extracellular matrix to image tumor progression, metastasis, and fibrosis.	[84]

* Alternatively spliced EIIIB domain of fibronectin.

**Table 2 cancers-16-00371-t002:** VHH application in radioimmunotherapy.

Targets	Conjugates	Cancer Models	References
HER2	^131^I (β/γ)	HER2^+^ cancer	[98,99]
	^125^I, ^131^I-SGMIB (β/γ)	BrCa	[100]
	^177^Lu (β)	OvCa	[94]
	^211^At (α)	HER2^+^ cancer	[101,102]
	^225^Ac (α)	SKOV3, BrCa	[103]
	^211^At (α)	BrCa	[102]
FAPα *	^89^Zr (γ), ^177^Lu(β)	FAPα^+^ cancer	[96]
	^131^I-SGMIB (β/γ)	FAPα^+^ cancer	[95]
	^225^Ac (α)		
MMR	^177^Lu (β), ^111^In(γ)	TS/A	[97]

* The anti-FAPα VHH is fused with the Fc fragment.

**Table 3 cancers-16-00371-t003:** Preclinical studies investigating VHH-PS conjugates.

Targets	Conjugates	Cancer Models	Main Findings	References
HER2	IRDye700DX	SK-BR-3 (HER2+, sensitive), HCC1954, JIMT1, HCC1419 (HER2+, resistant), MCF7 (HER2 low), MDA-MB-231 (HER2−)	Anti-HER2 VHH-PS could potently and selectively induce cell death in HER2-positive cells regardless of its sensitivity to trastuzumab.	[106]
EGFR	IRDye700DX	A431	The PS was conjugated with ^111^In-VHH in a site-specific way, which resulted in light-induced toxicity via cellular internalization.	[105]
EGFR	IRDye700DX	Cell lines with different EGFR expression	The anti-EGFR VHH-PS led to approx. 90% tumor necrosis and almost no toxicity in healthy tissue 24 h after PDT.	[107,108]
EGFR	IRDye700DX	A431, SCC-U8	VHH-PS induced the release of DAMPs (HSP70, ATP) and the pro-inflammatory cytokines of moDCs by incubating it with a conditioned medium, which stimulates the immune system.	[109]
MET	IRDye700DX	MKN45	The anti-MET VHH-PS had a nanomolar affinity and led to cell death at nanomolar concentration with illumination.	[110]
US28	IRDye700DX	U251-iUS28	The anti-US28 VHH-PS was the first example using GPCR as a target for VHH-directed PDT, which selectively killed US28-expressing glioblastoma cells.	[7,32]
EGFR/VEGFR2	IRDye700DX	OSCC	The dual-targeting VHH-PS showed improved efficacy in co-culture of endothelial and cancer cells.	[111]

**Table 4 cancers-16-00371-t004:** Preclinical studies using VHH-based ICIs.

Targets	Cancer Models	Main Findings	References
PD-L1+CD16a+IL15	PD-L1^+^ cancer	The fusion promoted cell growth in vitro, while it attenuated tumor growth in vivo.	[120]
PD-L1	PaCa	The VHH-CCL21 fusion could target PD-L1 positive TME and promote recruiting effector cells.	[121]
PD-L1	MC38	VHH outperformed conventional antibodies in inhibiting tumor growth due to VHH’s higher tumor penetration in the MC38 tumor.	[113]
PD-L1	PD-L1^+^ cancer	Monovalent, bivalent, and trivalent agents enhanced TCR signaling in PD-L1 positive cancer cells, to result in CD8^+^ T cell activation and cytokine production to attenuate cancer progression.	[122,123,124]
PD-L1+TIGIT	MC38	The multivalent bispecific VHH could synergistically enhance T cell activity by inhibiting tumor growth in vitro.	[116]
PD-1	A549, BxPC3	The VHH could block the PD-1/PD-L1 interaction.	[125]
PD-1	MC38	Long-term systemic expression of VHH by AAV vector provided anti-tumor activity without toxicity.	[126]
CTLA-4	Melanoma	The anti-CTLA4 VHH delayed melanoma growth and prolonged the survival time in mice.	[127]
CTLA-4	MC38, H22	The half-life-extended version of VHH exhibited therapeutic efficacy in a Fc independent manner.	[114]
4-1BB+PD-L1	CT-26-huPD-L1, MC-38-huPD-L1	The bispecific VHH showed anti-tumor efficacy with negligible hepatotoxicity.	[55]

**Table 5 cancers-16-00371-t005:** Preclinical studies evaluating the therapeutic efficacy of anti-TAAs VHHs in solid cancer.

Target	Cancer Models	Main Findings	Reference
CapG	MDA-MB-231	Anti-CapG VHH prevented the formation of lung metastasis.	[131]
CD38	Melanoma	Anti-CD38 VHH Pseudomonas exotoxin A (PE38) showed highly selective cytotoxicity. The effectiveness could be increased by retinoid acid.	[132]
CD47	Melanoma	Anti-CTLA4 VHH synergized with other immune therapies when CD47 in TME was near-completely blocked.	[133]
CEACAM5/CD3	LS174T, SKOV3	The in vivo half-life of the bispecific VHH was increased 12-fold via the PEGylation strategy, accompanied by more potent tumor inhibition.	[45]
CXCR7	HNSCC	The anti-CXCR7 VHH inhibited tumor growth by reducing the secretion of CXCL1 in vitro and inhibiting angiogenesis in vivo.	[134]
DLL4	MKN, HEK293	The DLL4 could bind on the surface of MKN cells, and gastric carcinoma tissue and inhibit the maturation of capillary-like structures in HUVECs.	[51]
DR5	Hela, Colo205	Multivalent anti-DR5 VHHs had higher apoptotic capacity than the monovalent form that could mimic the activity of the natural TRAIL ligand.	[135]
DPYSL2, TUFM, Vimentin, NAP1-L1	GBM	The anti-TUFM VHH showed a cytotoxic effect on GBM CSCs, while other VHHs were shown to target mature GBM cells.	[136]
EGFR	LUAD	VHH was linked with the cell-penetrating peptide nonaarginine. The VHH inhibited intracellular signaling by binding EGFR resulting in reduced cell migration.	[137]
EGFR	A549, DU145, MCF-7	The anti-EGFR extracellular domain III VHH showed an anti-tumor effect both in vitro and in vivo.	[138]
EGFR	SW480	VHH could inhibit cancer cell viability by altering proteins involved in the DNA-damage checkpoint process.	[139]
MET	HepG2, SK-HEP-1, HCC827, NIH3T3	Anti-MET VHH pool that acts against the whole ectodomain of MET could overcome MET targeted treatment resistance by promoting MET degradation and blocking the kinase activity of MET. The anti-MET VHH treatment could suppress cancer proliferation, viability, and colony formation in vitro and tumorigenesis in vivo.	[140]
p38δ	Hela	The VHH inhibited the target kinase activity and tumor growth.	[141]
Survivin	HepG2	The VHH targeted survivin and blocked the signaling pathway resulting in apoptosis.	[142]
Tie1	U87MG	Targeting Tie1 with specific VHH triggered Tie1-dependent inhibition of RTK phosphorylation and angiogenesis in endothelial cells and suppressed GBM viability and migration.	[143]

**Table 6 cancers-16-00371-t006:** Studies evaluating the therapeutic potential of VHH-drug conjugates.

Target	Cargo	Cancer Models	Main Findings	Reference
EGFR	Mal-Pt	A375, A431	The VHH-DC could be specifically internalized into EGFR-positive cancer cells, resulting in higher therapeutic effects and lower side effects compared with cisplatin alone.	[89]
PSMA	Doxorubicin	PC3-PIP, PC3-flu *	An in vivo study showed that a 42-fold lower amount of VHH-DC could result in similar tumor growth inhibition compared with commercial doxorubicin treatment.	[150]
HER2	Doxorubicin	BT474-M3, NCI-N87	VHH-DC could simultaneously bind with the HER2 target on cancer cells with trastuzumab, which results in synergistic antitumor activity.	[149]
HER2	Auristatin F	BT474, MDA-MB-231	VHH-DC-albumin fusion overcame the rapid renal clearance, which resulted in long-lasting tumor remission.	[47]
VEGFR2	Diphtheria Toxin	PC3	Coupling toxin with immune “carrier” resulted in cancer cell growth inhibition, while toxin alone was ineffective.	[151]
CD147	Doxorubicin	Hela, 4T1, U87, 293T(low), SMMC-7721	In vitro studies showed the VHH-DC could inhibit tumor cell proliferation and induce cell apoptosis. The VHH-DC had a synergistic effect in inhibiting the growth of tumors in vivo, as compared with the treatment of doxorubicin or VHH monotherapy.	[152]
5T4	SN38	BxPC-3, Huh-7	N501-SN38 showed deeper tumor penetration, higher tumor uptake, and faster accumulation at the tumor site than conventional ADC and exhibited effective antitumor activity both in vitro and in vivo.	[147]

* PC3-PIP and PC3-flu are PSMA positive or PSMA negative PC3, respectively.

**Table 7 cancers-16-00371-t007:** Overview of registered clinical trials of VHHs-based agents in the treatment of solid cancer.

Agent	Target	Cancer Type	Study Identifier	Phase	Status	Primary Purpose	Related Publication
^99m^Tc-NM-02	HER2	Breast cancer	NCT04040686	Early Phase I	Recruiting	Diagnostic	[177]
^99m^Tc-NM-01	PD-L1	Non-Small Cell Lung Cancer	NCT02978196	Early Phase I	Recruiting	Diagnostic	[178,179]
^99m^Tc-MIRC208	HER2	HER2 positive cancer	NCT04591652	Not Applicable	Recruiting	Diagnostic	[180]
^89^Zr-KN035	PD-L1	PD-L1 positive solid tumor	NCT04977128	Not Applicable	Recruiting	Diagnostic	[181]
^68^Ga-THP-APN09	PD-L1	Lung cancerMelanoma	NCT05156515	Not Applicable	Recruiting	Diagnostic	[181]
^68^Ga-NOTA-Anti-MMR-VHH2	MMR	Breast cancerHead and Neck cancerMelanoma (skin)	NCT04168528	Phase I/IIa	Recruiting	Diagnostic	[73,182,183]
^68^Ga-NOTA-Anti-MMR-VHH2	MMR	Breast cancerPancreatic cancerSalivary gland cancerGastric cancerEndometrial cancerUterine cancerNon-Small Cell Lung CancerBiliary tract cancerCholangiocarcinomaColorectal cancerUrothelial carcinomaProstate cancer	NCT03924466	Phase II	Recruiting	Diagnostic	[73,182,183]
^68^Ga-NOTA-Anti-HER2 VHH1	HER2	Breast cancer	NCT03924466	Phase II	Recruiting	Diagnostic	[18,64,184]
^68^Ga-NOTA-Anti-HER2 VHH1	HER2	Breast cancer	NCT03331601	Phase II	Recruiting	Diagnostic	[18,64,184]
^99m^Tc-NM01	PD-L1	Non-Small Cell Lung Cancer	NCT04992715	Phase II	Recruiting	Diagnostic	[178,179]
^131^I-SGMIB Anti-HER2 VHH1	HER2	Breast cancer	NCT02683083	Phase I	Completed	Diagnostic	[185]
^68^Ga-ACN376	CLDN18.2	Solid tumor	NCT05436093	Not Applicable	Recruiting	Screening	
αPD1-MSLN-CAR-T Cells	PD-1	Solid tumor	NCT05373147	Early Phase I	Recruiting	Treatment	[186]
αPD1-MSLN-CAR-T Cells	PD-1	Colorectal cancerOvarian cancer	NCT04503980	Early Phase I	Recruiting	Treatment	[186]
αPD1-MSLN-CAR-T Cells	PD-1	Non-small-cell Lung CancerMesothelioma	NCT04489862	Early Phase I	Recruiting	Treatment	[186]
αPD1-MSLN-CAR-T Cells	PD-1	Colorectal cancer	NCT05089266	Phase I	Not yet recruiting	Treatment	[186]
KN046+Axitinib	PD-L1/CTLA4 Bispecific	Advanced Non-small Cell Lung cancer	NCT05420220	Phase II	Not yet recruiting	Treatment	[187]
KN046	PD-L1/CTLA4 Bispecific	Thymic carcinoma	NCT04469725	Phase II	Recruiting	Treatment	[188]
KN044	CTLA4	Advanced solid tumor	NCT04126590	Phase I	Recruiting	Treatment	[189]
KN035	PD-L1	Solid tumor	NCT03101488	Phase I	Completed	Treatment	[190,191]
KN035	PD-L1	Advanced or metastatic solid tumor	NCT03248843	Phase I	Completed	Treatment	[190,191]
JS014 (fusion with IL-21) + Pembrolizumab	Human Serum Albumin	Malignant neoplasm Experimental solid tumorAdult lymphoma	NCT05296772	Phase I	Active, not recruiting	Treatment	
Gavocabtagene autoleucel (gavo-cel; TC-210)	Mesothelin	Mesothelioma	NCT03907852	Phase I Phase II	Recruiting	Treatment	[192]
Envofolimab (KN035)+Gemcitabine and Cisplatin	PD-L1	Biliary tract cancer	NCT04910386	Phase II	Not yet recruiting	Treatment	[193]
^99m^Tc-NM-02, ^188^Re-NM-02	HER2	Breast cancer	NCT04674722	Early Phase I	Recruiting	Treatment	[177,194]
Envafolimab (+Ipilimumab)	PD-L1	Pleomorphic sarcoma Myxofibrosarcoma	NCT04480502	Phase II	Recruiting	Treatment	[195]
^68^Ga-NODAGA-SNA006	CD8α	Solid tumors	NCT05126927	Early Phase I	Recruiting	Diagnostic	[196]
DR30303-IgG1Fc	CLDN18.2	Malignant neoplasm of the digestive system	NCT05639153	Phase I	Recruiting	Treatment	[197]
[^99^mTc]-NM-01	PD-L1	Non-small cell lung cancer, malignant melanoma	NCT04436406	Not Applicable	Recruiting	Diagnostic	
^68^Ga-PD-L2	PD-L2	Colorectal cancer, Lung cancer	NCT05803746	Not Applicable	Recruiting	Diagnostic

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
