# Peer review of "Promising Diagnostic and Therapeutic Approaches Based on VHHs for Cancer Management"

_cancers, 2024, doi:10.3390/cancers16020371_

Round 1

Reviewer 1 Report

Comments and Suggestions for Authors

In this article  Ying Cong et al. present an interesting current review of single-domain antibodies originally described  in species belonging to the Camelidae family, also known as VHH or nanobody, with applications in tumor imaging and cancer therapy.

This review, provide general characteristics of VHH: Biochemical and Biophysical Characteristics such as high affinity. high tissue penetration , fast blood clearance.  and low immunogenicity (the relatively small size of VHH and the lack of Fc fragment which prevents one of the main  causes of adverse effects in antibodies therapies. The general route of VHH  development. Applications like: VHH-based imaging, VHH as Immune Checkpoint Inhibitor, VHH to  neutralize cytokines or growth factors such as TNFα or VEGF, attenuating tumor growth by inhibiting metastasis or angiogénesis. Furthermore, they summarize the applications of VHH in anti-cancer therapies promising options for improving outcomes of cancer patients and finally, the review provide an interesting overview of registered clinical trials testing VHH-based therapeutic  and diagnostic agents for oncological diseases. Most of VHHs in ongoing clinical trials are still at their early stages. (data are recents; Collected on ClinicalTrial.gov on 24th Aug. 2023).

Author Response

As attachment 

Reviewer 2 Report

Comments and Suggestions for Authors

In the manuscript,  Promising Diagnostic and Therapeutic Approaches Based on VHHs for Cancer Managements,” Conng and co-workers reviewed the characteristics, advantages, disadvantages, and applications of 19 VHHs compared to conventional antibodies. However, some key points that need further discussion are listed below:

Comments

·      -The review should have one or two more figures that schematically represent the mechanisms of how VHH antibodies act and what signaling pathways they induce or repress. This would make the review more interesting and not just have tables.

·    -The manuscript requires a revision style of the English language. Furthermore, some paragraphs are difficult to understand.

Comments on the Quality of English Language

The manuscript requires a revision style of the English language. Furthermore, some paragraphs are difficult to understand.

Author Response

As attachment 

Reviewer 3 Report

Comments and Suggestions for Authors

In this interesting manuscript, Cong and colleagues have collected evidences within VHHs application in cancer diagnosis and therapy, hence described nanobodies characteristics and development.

The review is properly designed and tables are clear, yet a double check of few elements should be done and minor adjustements could improve the paper.

In particular, please double check the punctuation in the title and the references; furthermore, the numbering and references of table 7, and the subchapters in chapter 3.

Author Response

As attachment 
